# IN THE KNOWN, OUT OF THE ORDINARY: PROBING OOD DETECTION WITH SYNTHETIC DATASETS

## ABSTRACT

Out-of-distribution (OOD) detection is crucial for ensuring the reliability of machine learning models, especially in visual tasks. Most existing benchmarks focus on isolating distribution shifts and creating varying levels of detection difficulty, often relying on manual curation or classifier-based scoring with human annotations. Additionally, large-scale benchmarks are typically derivatives of ImageNet-21k classes or combinations of ImageNet with other datasets. However, no existing work offers a setup where only one attribute such as color or class changes in a controlled manner, while other attributes of the object remain constant. This limits our ability to precisely study the impact of individual attributes on OOD detection performance. We aim to address this by proposing two novel synthetic datasets, SHAPES and CHARS, designed to explore OOD detection under controlled and fine-grained distribution shifts. SHAPES consist of 2D and 3D geometric shapes with variations in color, size, position, and rotation, while CHARS consists of alphanumeric characters with similar variations. Each dataset presents three scenarios: (1) known classes with unseen attributes, (2) unseen classes with known attributes, and (3) entirely novel classes and attributes. We train 10 architectures and assess 13 OOD detection methods across the three scenarios, concentrating on the impact of attribute shifts on OOD scores, while also conducting additional analysis on how image corruption influences OOD scores. By systematically examining how specific attribute shifts affect OOD scores and the affects of noisy test samples, we aim to bring greater transparency to where these methods succeed or fail, helping to identify their limitations under various conditions.

## 1 INTRODUCTION

Out-of-distribution (OOD) detection is crucial for ensuring the reliability of machine learning models in real-world applications. While models perform well on in-distribution (ID) data, they often fail on unseen OOD inputs, providing high-confidence predictions despite being wrong (Amodei et al., 2016). OOD detection methods mitigate this by identifying unfamiliar data and prevent incorrect predictions, which is vital in high-stakes areas such as healthcare, autonomous systems, and security. Recent advancements in OOD detection encompass a variety of approaches, including classification-based methods, density-based models, and distance-based techniques (Yang et al., 2024a).

Initially, OOD detection methods were evaluated using small-scale datasets with relatively simple in-distribution (ID) and out-of-distribution (OOD) pairs. For instance, CIFAR-10 and CIFAR-100 (Krizhevsky, 2009) were commonly used as ID datasets, while OOD data included datasets such as SVHN (Netzer et al., 2011), LSUN (Yu et al., 2015), Places365 (Zhou et al., 2018), and Textures (Cimpoi et al., 2014). Later, larger benchmarks began incorporating more complex and diverse datasets to better reflect real-world distribution shifts. ImageNet1k (Deng et al., 2009) became a standard ID dataset and the corresponding OOD datasets included iNaturalist (Van Horn et al., 2018) and classes from ImageNet21k (Ridnik et al., 2021) which were not present in the ID dataset.

Recent works in benchmarking OOD methods has focused on overcoming limitations of fixed ID-OOD dataset pairs. Datasets such as OpenImage-O (Wang et al., 2022), ImageNet-OOD (Yang et al., 2024b) and C-OOD (Galil et al., 2023) provide more natural, diverse, and scalable benchmarks, addressing issues such as predefined class overlaps, limited coverage, and covariate contamination.

Nevertheless, the field still lacks a framework that provides precise control over individual attributes, which is essential for gaining deeper insights into the reasons behind the success or failure of OOD detection methods.

**Our Contributions:** To address this gap, we introduce a synthetic approach involving two carefully designed datasets, SHAPES and CHARS. SHAPES consists of simple 2D and 3D geometric primitives such as squares, cubes, and spheres, while the dataset CHARS contains alphanumerical characters. Each dataset presents test sets where specific attributes of the images are systematically varied. For simplicity, we focus on three controlled scenarios: (1) Known classes with unseen attributes, where we modify the color of the objects while keeping the class from the training distribution constant—this setup represents a covariate shift; (2) Unseen classes with known attributes, where the color remains unchanged, but the object class is new to the model—this setup resembles a semantic shift with visual similarity to the training data; and (3) Entirely novel classes and attributes, where both the class and color of the object are completely unfamiliar to the model. We also introduce image corruption in test sets to study how OOD methods respond to noisy inputs. By examining how OOD methods respond to controlled distribution shifts and studying their score behavior in the presence of corrupted test samples, we aim to provide deeper insights into the conditions that cause these methods to fail and assess their resilience to minor perturbations, such as noise or distortions.

## 2 SHAPES AND CHARS

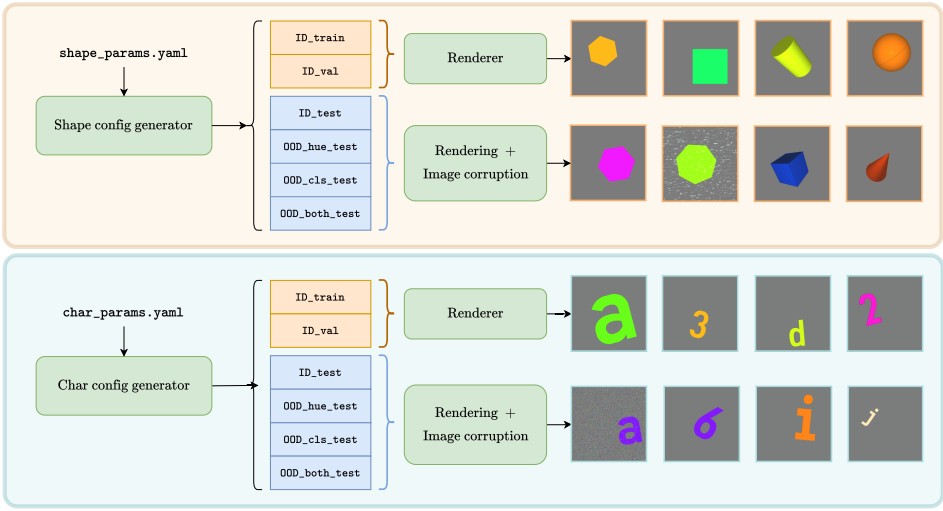

Figure 1: Dataset generation pipeline for SHAPES and CHARS datasets.

We introduce two new synthetic datasets, SHAPES and CHARS. SHAPES includes basic 2D and 3D geometric primitives, such as squares, cubes, and spheres, whereas the CHARS dataset comprises alphanumeric characters. To create samples in both datasets, a consistent process is followed. Each dataset has its own configurations specifying attributes such as color, rotation, size range, and other dataset-specific properties. These configurations also define the number of samples for train, validation, ID (in-distribution) test, and 3 OOD (out-of-distribution) splits (see Figure 1). The attribute values for ID and OOD are disjoint sets, ensuring clear separation between in-distribution and out-of-distribution samples. The background color for all images remains the same across all samples.

The three OOD splits are defined as follows:

- *OOD in color*: Images have their colors sampled from OOD colors, while their classes remain a subset of ID classes.

- *OOD in class*: Images use ID colors but belong to completely unseen classes not present in the training set.
- *OOD across both*: Configurations are generated by sampling entirely from OOD attributes, meaning both class and color are OOD.

The configuration generator pre-generates all configurations required for train, validation, and test splits with a fixed random seed, ensuring reproducibility and consistency. Images are rendered dynamically in the dataloader, which fetches the necessary configurations for each batch and renders the images using moderngl (Dombi, 2020). For test splits, a specified percentage of images, as defined in the configuration, are pre-assigned with corruption details during the configuration generation phase. The corruption details include a corruption method and a severity level (either 1 or 2), selected from one of the ten common image corruption strategies mentioned in Hendrycks & Dietterich (2019). For a detailed overview of the exact attributes and values used, refer to the Appendix A.

# 3 EXPERIMENTS AND ANALYSIS

## 3.1 EXPERIMENTAL SETUP

**Problem Setup:**. Let $\mathcal{D}_{\text{in}} = \{(x_i, y_i); x_i \in \mathcal{X}, y_i \in \mathcal{Y}\}$ represent the In-distribution data (i.e., data the model is trained on) sampled from distribution $P_{\text{in}}(x, y)$, where $x \in \mathbb{R}^d$ is the input image and $y$ is the corresponding label. The Out-of-distribution data $\mathcal{D}_{\text{out}}$ comes from a different distribution $P_{\text{out}}(x, y)$ which is not seen during training.

Given a classifier $f : \mathbb{R}^d \to \mathbb{R}^N$ trained on $\mathcal{D}_{\text{in}}$ that classifies input data to $N$ In-distribution classes, the goal of Out-of-distribution detection is to design a scoring function $S(x)$ that helps in distinguishing between in-distribution data $\mathcal{D}_{\text{in}}$ and out-of-distribution data $\mathcal{D}_{\text{out}}$. The decision is made on a threshold $\tau$, where:

$$S(x) = \begin{cases} \text{In-Distribution}, & \text{if } S(x) \geq \tau, \\ \text{Out-of-Distribution}, & \text{if } S(x) < \tau. \end{cases}$$

We evaluate the OOD methods under three types of OOD scenarios: 1) where only the color of the shape/character changes 2) where only the class changes, and 3) where both color and class change. We will use the terms 'OOD in color,' 'OOD in class,' and 'OOD in both' as shorthand for addressing these OOD types.

**Datasets:** We prepare the SHAPES and CHARS datasets by setting a random seed to generate image configurations for training, validation, ID test and three OOD test splits (OOD in color, class, and both). Across all four test splits (one in-distribution and three OOD) in both datasets, A portion of the images is corrupted using one of ten corruption methods applied to each image at a specific severity level (either 1 or 2). To ensure consistency, we repeat the training and evaluations using three random seeds.

**Backbones:** We select 10 architectures combined across the ResNet (He et al., 2016), DenseNet (Huang et al., 2017), Vision Transformer (ViT) (Dosovitskiy et al., 2020), and Wide-ResNet (Zagoruyko & Komodakis, 2016) families, each with a single linear layer as the classification head. The output dimension of the classification head corresponds to the number of in-distribution (ID) classes for each dataset. All models are trained independently from scratch on both datasets.

**OOD methods:** We evaluate 13 OOD detection methods comprising of logit-based, feature-based and energy-based methods across the three OOD scenarios. Logit-based methods include ODIN (Liang et al., 2018), MaxLogit (Hendrycks et al., 2022), MSP (Hendrycks & Gimpel, 2017), and ViM (Wang et al., 2022), all of which operate directly on logits or modify them to compute OOD scores. Feature-based methods include SCALE (Xu et al., 2024), SHE (Zhang et al., 2023b), GradNorm (Huang et al., 2021), KNN (Sun et al., 2022), and NNGuide (Park et al., 2023), which work on feature representations, typically from the penultimate layer. Lastly, energy-based methods, include EBO (Liu et al., 2020), GEN (Liu et al., 2023), ASH Djurisic et al. (2023), and ReAct (Sun et al., 2021), which calculate an energy score derived from logits or modified activation's. All OOD detection methods are implemented using the OpenOOD framework laid out by Zhang et al. (2023a).

**Evaluation Metrics:** We evaluate the OOD-detection performance using the commonly used metric AUROC. It represents the probability that a positive example receives a higher detection score than a negative example (Fawcett, 2006). Higher value indicates better detection performance. AUROC values are calculated for each dataset, for each type of OOD scenario, both with and without image corruption. The reported AUROCs correspond to the median AUROC across the three seeds. The observed absolute deviation from the median (MAD) for AUROC across the three seeds for all OOD methods and backbone combinations was in the order of $10^{-2}$.

We use the *Overlap Coefficient* (eq-1) to measure the overlap between the smoothed densities of min-max normalized OOD scores of ID and OOD test samples.

$$\text{overlap}(A, B) = |A \cap B|, \quad 0 \le \text{overlap}(A, B) \le 1. \tag{1}$$

The set notations used in Equation 1 are for the sake of brevity.

### 3.2 SENSITIVITY OF OOD METHODS TO COLOR

Figure 2 presents the AUROC scores for un-corrupted test images across all combinations of OOD methods and architectures on both datasets in two OOD scenarios: OOD in color and OOD in class. Except for methods such as KNN, React, and ViM, other methods perform poorly, with AUROC as low as 0.01, which is worse than random coin flip. The reason for this can be seen in Figure 3, where OOD samples are given higher scores than ID samples in 'OOD in color' scenario. This observation is quite opposite to the intended behavior of OOD detection methods, where ID samples should have received higher scores than OOD samples.

The AUROC values in the scenario when both color and class change, is nearly identical to that of the 'OOD in color' scenario and the results are presented in Figure 5 in the Appendix. It can again be seen in Figure 3, where the score distributions for OOD in color and OOD in both color and class scenarios remain similar. This further reinforces the evidence that OOD detection methods are highly sensitive to changes in visual attributes like color. Further among the selected architectures, we observe that ViT performs the best across all OOD methods and amongst the 13 OOD methods, KNN and ViM are robust across all the architectures (Figure 2).

### 3.3 IMPACT OF IMAGE CORRUPTION

To assess the impact of image corruption on OOD scores for both ID and OOD samples, we first extract the OOD scores for corrupted ID and OOD test sets from all OOD methods, across all architectures, and apply min-max normalization such that the relative order of score distributions and scales are preserved. Then using the overlap coefficient 1, we measure the overlap between the smoothed densities of normalized OOD scores of ID and OOD test samples. Figure 4 shows the histogram of 130 such overlap coefficients obtained by all the OOD method and backbone combinations. Intuitively, when there is no corruption, the overlap coefficients across all the OOD methods and architectures should be relatively lower, indicating the ability of OOD methods to assign a higher score to the ID samples. But with image corruption, one might expect a higher overlap given the poor performance of OOD methods in Figure 2. Henceforth pointing towards a conjecture that, an ID corrupted image is as bad as an OOD image (with or without corruption). We precisely corroborate this intuition in Figure 4, where we find that overlap coefficients across the OOD method and backbone combinations increase in the presence of image corruption and significant in the case of OOD in class.

The AUROC plots for corrupted images across all OOD methods and architectures are provided in Figure 6 in the Appendix. As seen in Figure 2, KNN and ViM remain robust OOD methods and ViT, the best amongst the chosen architectures. Though we can observe a slight increase in the AUROC for OOD in color and OOD in both color and class cases (relative to Figure 2 and Figure 5 respectively), there is a decrease in AUROC for OOD in color scenario. These observations can be attributed to the same line of observation we made in Figure 4 that an ID corrupted image is as bad as an OOD image, which inflates or shrinks the AUROC values which suit a random coin toss.

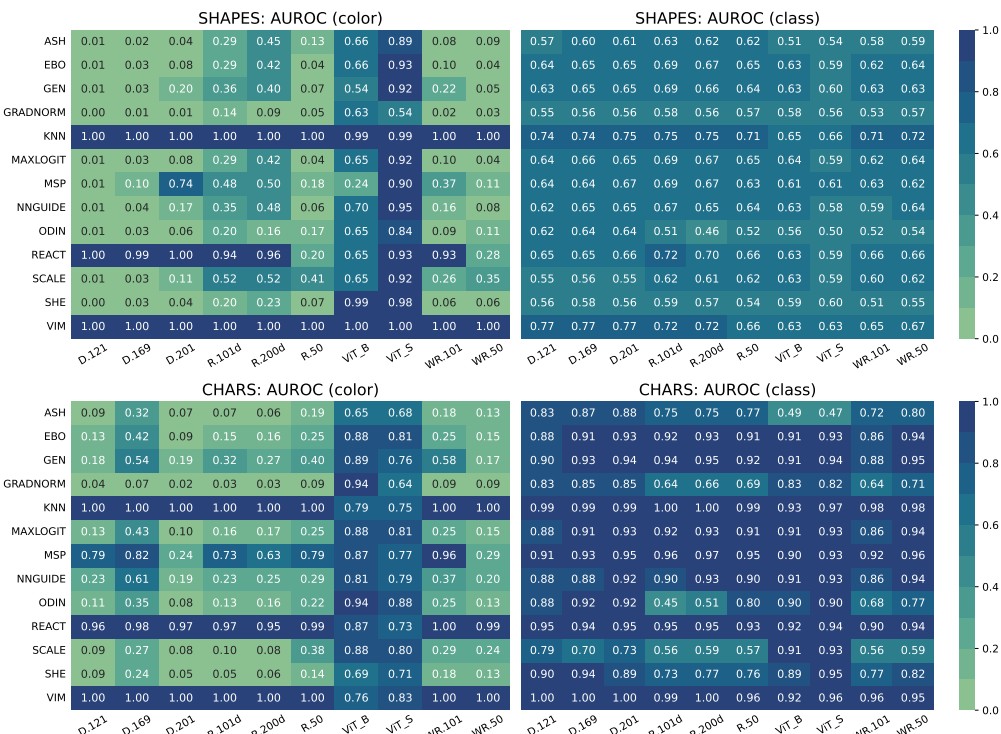

Figure 2: AUROC of OOD detection methods across all architectures on uncorrupted test images, comparing two OOD scenarios: OOD in color (left column) and OOD in class (right column), for datasets SHAPES (top row) and CHARS (bottom row). Model abbreviations: **D**: DenseNet, **R**: ResNet, **ViT**, and **WR**: Wide ResNet.

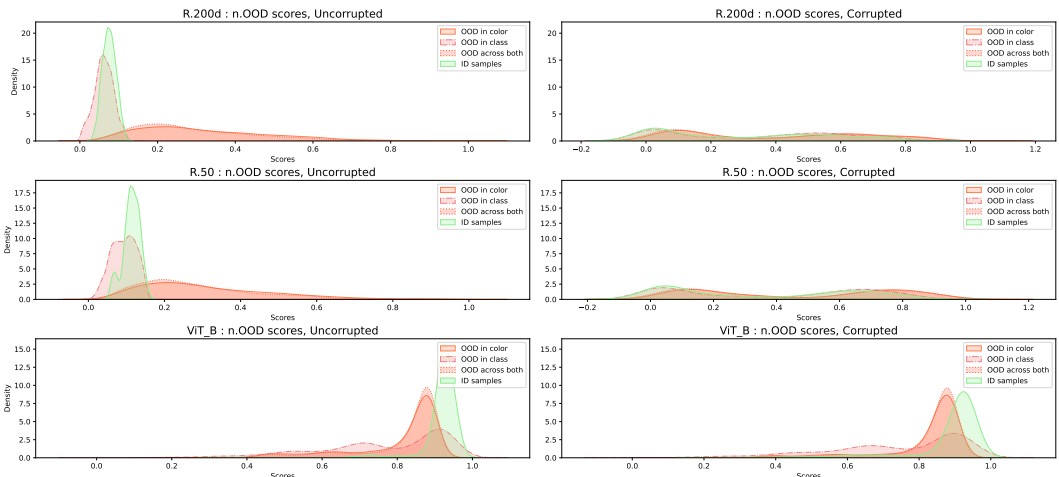

Figure 3: Normalized OOD score distributions for ID and OOD images using the GradNorm OOD method on the **CHARS** dataset. The left column shows un-corrupted images, while the right shows corrupted images. This representative example illustrates ID samples receiving lower scores than OOD samples, a pattern consistent across various OOD methods and architectures with similarly low AUROC scores. Model abbreviations: **R**: ResNet and **ViT**.

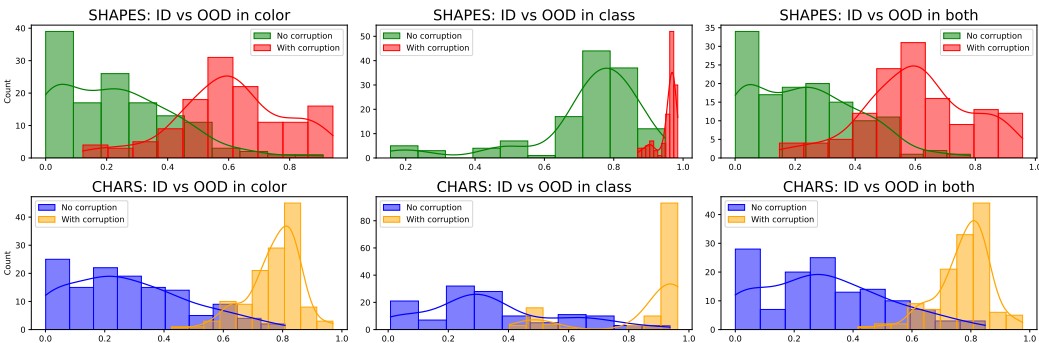

Figure 4: Histograms of overlap coefficients between ID and OOD score distributions, measured across all combinations of OOD methods and models, with two rows for the datasets SHAPES(top) and CHARS(bottom), with each row containing 3 subplots for different OOD test types: color, shape, and both. Each subplot compares overlap coefficients with and without image corruption.

## 4 RELATED WORK

**Trends in the evaluation of OOD Methods:** The evaluation of out-of-distribution (OOD) detection methods has evolved significantly over time. Initially, OOD detection methods were evaluated on simpler, small-scale datasets with low-resolution images. Common choices for in-distribution (ID) datasets during this early phase included CIFAR-10, CIFAR-100 (Krizhevsky, 2009), and SVHN (Netzer et al., 2011). As for the out-of-distribution (OOD) datasets, selections were often visually distinct and low-resolution datasets such as LSUN (Yu et al., 2015) (Crop and Resize), Places365 (Zhou et al., 2018) and Textures (Cimpoi et al., 2014). While these dataset pairs offered some insight into OOD detection performance, their limitations became increasingly apparent. The ID and OOD datasets were typically quite different in terms of both visual appearance and resolution, often leading to an overestimation of OOD detection performance, and failed to reflect real-world distribution shifts encountered in more complex domains. Recognizing these limitations, more recent methods such as ViM (Wang et al., 2022) and NNGuide (Park et al., 2023) shifted toward using ImageNet-1k (Deng et al., 2009) as the ID dataset, introducing more realistic scenarios for OOD detection. This shift also brought about the adoption of larger, more challenging OOD datasets such as subsets of ImageNet-21k (Ridnik et al., 2021) and iNaturalist (Van Horn et al., 2018).

**Existing OOD Benchmarks:** Datasets such as OpenImage-O (Wang et al., 2022) were developed to overcome problems such as OOD datasets relying on predefined class labels, which can overlap with in-distribution (ID) classes and offer limited coverage. OpenImage-O provides more diverse and realistic OOD examples. Similarly, ImageNet-OOD (Yang et al., 2024b) focuses on reducing covariate shifts and resolving semantic ambiguity by selecting OOD classes that do not overlap with ImageNet-1K, allowing for a more targeted evaluation of semantic shifts. ImageNet-O (Hendrycks et al., 2021), on the other hand, addresses models' failures to detect OOD data by using adversarial filtering to stress-test models' high-confidence misclassifications. Other benchmarks, such as NINCO (Bitterwolf et al., 2023), tackle the contamination of OOD samples with ID examples, providing a cleaner and more diverse dataset for OOD evaluation. C-OOD (Galil et al., 2023) introduced a versatile framework for evaluating OOD detection across varying levels of difficulty, addressing the biases of earlier benchmarks.

While these efforts have significantly advanced the field, most of the focus was on improving the quality of a specific type of attribute shift or developing benchmarks with varying OOD difficulty levels. However, there is still a lack of understanding of how OOD methods perform when individual image characteristics, such as color or class, are changed, which we have investigated in this manuscript.

## 5 CONCLUSION

We present two novel synthetic datasets, SHAPES and CHARS, designed to explore the complexities of out-of-distribution (OOD) detection under controlled attribute shifts. By isolating variables such as color and class, these datasets allow for precise evaluation of how different OOD detection methods perform when encountering unseen data. The results highlight the sensitivity of OOD detection methods, particularly to changes in visual attributes such as color, and demonstrate that existing methods often struggle with fine-grained shifts in distribution. Furthermore, the introduction of image corruption as an additional challenge provides deeper insights into the robustness of these models. The findings suggest the need for continued refinement in OOD detection techniques to ensure reliability in real-world applications.

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

# A  IMAGE ATTRIBUTE SPECIFICATIONS

## A.1  COLOR

All colors are specified in HSV format. For both the SHAPES and CHARS datasets, the background color is fixed at [0, 0, 155] in HSV. Additionally, both datasets share the same set of in-distribution and out-of-distribution colors, which are:

```
color:
  id_hues: [30, 45, 60, 75, 90, 105, 120, 135, 150]
  ood_hues: [210, 225, 240, 255, 270, 285, 300, 315, 330]
  bg_color: [0,0,155]
```

## A.2  CLASSES

We use two types of classes in each of the datasets (SHAPES and CHARS): In-Distribution (ID) classes, which are used to for training the model, and Out-of-Distribution (OOD) classes, which differ entirely from the ID classes and are used for creating test sets.

The SHAPES dataset contains 17 classes, with 8 ID classes and 9 OOD classes. The ID and OOD classes are chosen to be conceptually related but distinct. For example, if the circle is in ID, then the ellipse is in OOD; if the cube is in ID, the cuboid is in OOD; similarly, if the square is in ID, the rectangle is in OOD.

```
shape:
  # rp = regular polygon
  id_shapes: [circle, square, rp6,
              rp8, eq_triangle, sphere_3d,
              cube_3d, cylinder_3d]
```

```
     ood_shapes: [ellipse, rectangle, rp7, rp9,
              is_triangle, random, ellipsoid_3d,
              cuboid_3d, cone_3d]
```

The CHARS dataset has 20 classes, a mix of alphanumeric characters. The first 5 alphabets and first 5 whole numbers are in ID, while the next 5 alphabets and numbers are in OOD. Since these are glyphs, we need a font to render the characters, and for our experiments, we use the *Monofonto-Regular* font.

```
  chars:
    id_chars: [a, b, c, d, e, 0, 1, 2, 3, 4]
    ood_chars: [f, g, h, i, j, 5, 6, 7, 8, 9]
```

### A.3 OBJECT SIZE, ROTATION, AND ADDITIONAL PARAMETERS

We define size bounds for SHAPES and font sizes for CHARS. The size bounds are specified as a range $[a, b]$, representing the minimum and maximum percentages of the image dimension. These values apply to size attributes such as the side lengths for polygons or the diameter for circles and ellipses. For CHARS, the font size also ranges between a minimum and maximum value. The size bounds and font sizes chosen are:

```
  size_bounds: [35,55]
  font_size_min_max: [60,150]
```

Both datasets allow for rotation within a specified range $[r_a, r_b]$. Additionally, we use a rotation angle step $s$, meaning that valid rotation angles are $r_a$, $r_a + s$, $r_a + 2s$, $\cdots$, $r_b$. 2D shapes and characters rotate only in the XY-plane, either clockwise or counterclockwise, while 3D objects can rotate along all three axes.

```
  shapes:
    rot_min_max_2d: [-180,180]
    rot_min_max_3d: [-60,60]
    step_angle: 10

  chars:
    rot_min_max: [-60,60]
    step_angle: 5
```

In some cases, additional information is needed to generate synthetic images. For instance, to create a random shape, we require parameters such as the number of points and smoothness. For 3D shapes, additional attributes, such as Phong lighting settings, are necessary for rendering.

```
  shape:
    rnd_shape:
      num_points: [5,6,7,8,9,10,11,12]
      min_smoothness: 30
      max_smoothness: 80
      min_radius_mult: 0.7
      max_radius_mult: 1.3

    solid_shape_params:
      ambient_strength_bounds: [0.25, 0.5]
      specular_strength_bounds: [0.15, 0.3]

      # offset of light from camera, in
      # camera plane for 3D shapes.

      light_pos_offset: 80
```

## A.4 Dataset Configuration and Split Details

As mentioned earlier in the main, we use three seeds (1, 2, and 3) to generate the dataset configurations. For both datasets, the number of images in the training, validation, and test sets (ID and OOD) are same.

```
imgs_per_split:
  train: 100000
  val: 5000

  test:
    id: 5000
    ood_hue: 5000
    ood_cls: 5000
    ood_both: 5000
```

## A.5 Image Corruption methods

We set `crrp_ratio` to 0.3 for both the datasets, indicating the percentage of images in each test split to be corrupted. In total, we apply 10 different corruption strategies, each with two severity levels. The severity values vary by method. For example, in `gaussian_noise`, severity is determined by the scale parameter, which represents the standard deviation of the distribution. A higher scale results in a blurrier image. The following are the 10 chosen corruption strategies applied in our evaluation:

```
corruption_methods = [
    "gaussian_noise",
    "shot_noise",
    "impulse_noise",
    "speckle_noise",
    "gaussian_blur",
    "glass_blur",
    "spatter",
    "contrast",
    "brightness",
    "saturate",
]
```

## A.6 Rendering Process

This is a simplified overview of the entire process, from generating image configurations to rendering the final images.

1: **Input:** Dataset attributes, and a random seed for reproducibility
2: **Output:** Configuration files for dataset splits, Rendered images
3: **Step 1:** Read image attribute configurations and set the random seed.
4: **Step 2:** For each image:
  - Sample attributes such as rotation, class, color, size, and other additional parameters using appropriate random sampling methods (e.g., `random.choices`)
  - Calculate margins for movement in X and Y directions based on the size and rotation of the image.
  - Sample random offsets in the X and Y directions from center, to place the shape/character.
5: **Step 3:** If the image belongs to the OOD test split:
  - Add additional OOD-related information, such as the type of OOD and image corruption strategies, to the configuration.
6: **Step 4:** Save all configuration files for each dataset split (Train, Validation, ID Test, OOD Test). These files will be used in the rendering process.
7: **Rendering Process**:

- Using `moderngl`'s headless contexts, configurations from dataloader's collate function are sent to respective graphical contexts.
- Images are rendered based on these configurations and are returned as batches.

# B SUPPLEMENTARY AUROC PLOTS

The AUROC results presented in Figure 5, where both the color and class of the objects are unseen, closely resemble the results obtained when only the color is altered, for the un-corrupted images. In many cases, the AUROC values are nearly identical or exactly the same, highlighting that changes in color have a significant influence on the scores produced by various OOD detection methods, while changes in the object's class appear to have a much smaller effect.

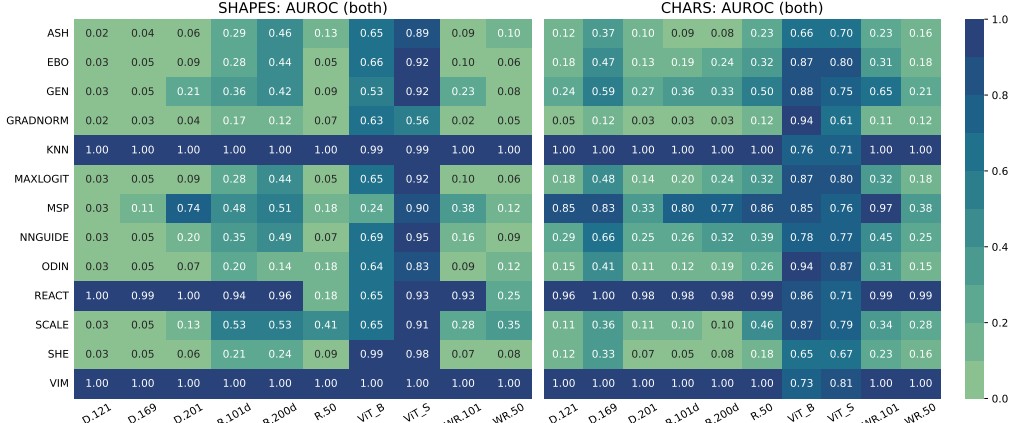

Figure 5: AUROC of OOD detection methods across all models on un-corrupted test images, where samples are OOD in both color and class, for the SHAPES and CHARS datasets. Model abbreviations: **D**: DenseNet, **R**: ResNet, **ViT**, and **WR**: Wide ResNet.

Figure 6 shows the AUROC values across all three cases for both datasets, with test images corrupted by one of the 10 corruption methods at varying severity levels. The impact of color remains significant, much like in the uncorrupted case. Most AUROC values across the different combinations perform poorly, often comparable to a random coin toss or even worse. However, as in the uncorrupted case, methods such as KNN and ViM stand out, showing better performance. ViT-based models also outperform other architectures and OOD detection methods in many cases.

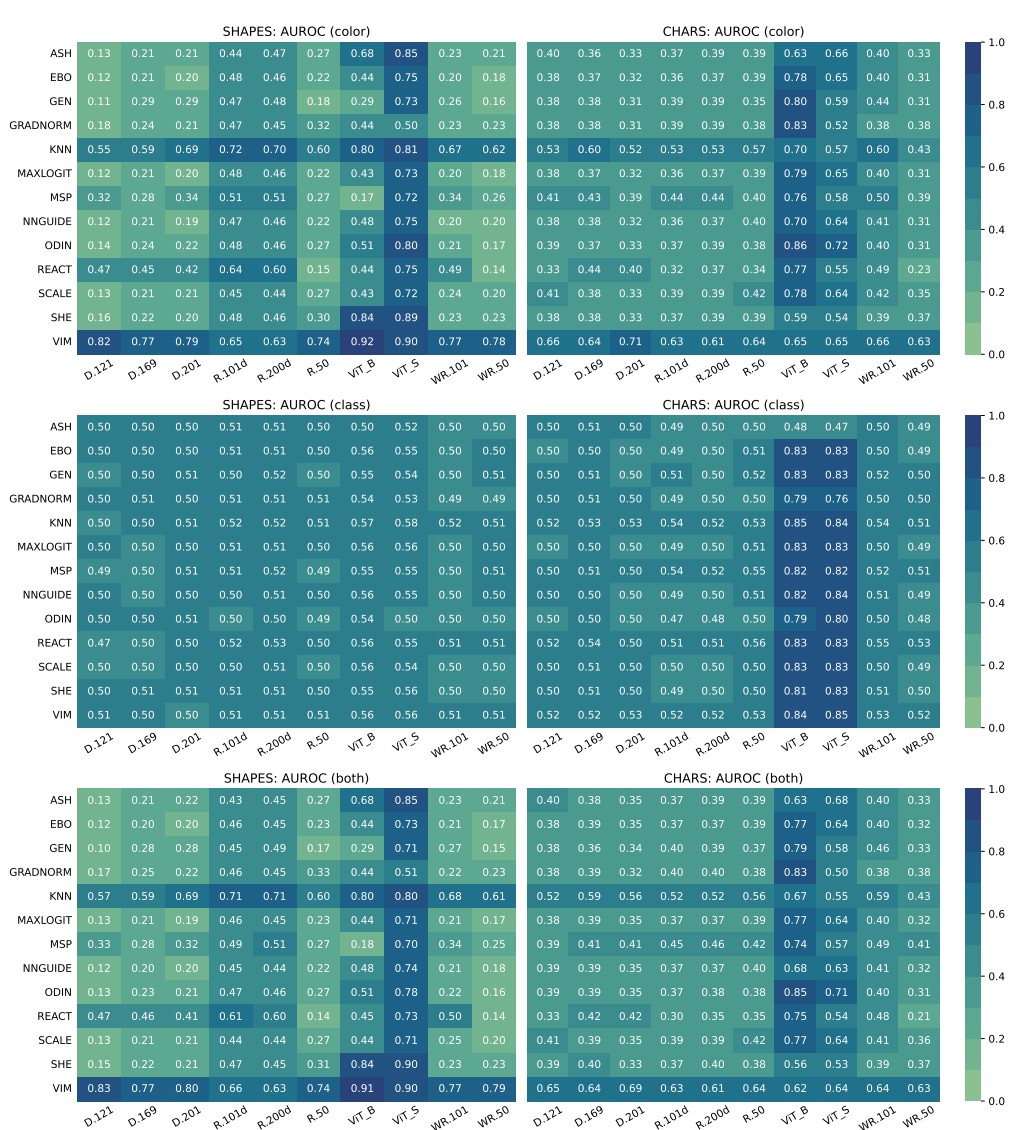

Figure 6: AUROC of OOD detection methods across all models on *corrupted* test images, comparing three OOD scenarios: OOD in color, OOD in class, and OOD in both color and class. Model abbreviations: **D**: DenseNet, **R**: ResNet, **ViT**, and **WR**: Wide ResNet.

