# OpenReview forum: "IN the known, OUT of the ordinary: Probing OOD detection methods with Synthetic datasets."
_ICLR.cc/2025/Conference — ICLR 2025 Conference Withdrawn Submission_

### Official Review · Reviewer_TB9J · 2024-10-28

**Soundness:** 1
**Presentation:** 1
**Contribution:** 1
**Rating:** 1
**Confidence:** 4

**Summary:**

This paper proposes two novel synthetic datasets, SHAPES and CHARS, designed to explore OOD detection under controlled and fine-grained distribution shifts.

**Strengths:**

This paper proposes two datasets.

**Weaknesses:**

I think this paper lacks rigorous theoretical analysis and sufficient background research. It should not be submitted as an academic paper to a top conference like ICLR, but more like a college student's course assignment that was submitted incorrectly.

**Questions:**

NA

---

### Official Review · Reviewer_xwHN · 2024-10-29

**Soundness:** 2
**Presentation:** 2
**Contribution:** 2
**Rating:** 3
**Confidence:** 4

**Summary:**

This paper explores out-of-distribution (OOD) detection by introducing two synthetic datasets, SHAPES and CHARS, designed to study controlled, fine-grained distribution shifts. Unlike prior benchmarks, these datasets isolate specific attributes—such as color and class—allowing for targeted analysis of OOD performance across various architectural and methodological approaches. The experiments focus on three OOD scenarios, including known classes with unseen attributes, unseen classes with known attributes, and entirely novel combinations, while also examining image corruption effects on OOD scores.

**Strengths:**

1. Originality: The paper’s approach to controlled OOD detection using attribute-focused datasets is unique, allowing for precise examination of how specific visual properties affect model robustness.
2. Quality: The experimentation is thorough, covering multiple architectures and OOD methods to provide a comprehensive overview of performance across different conditions.
3. Clarity: The structure and flow of the paper make the methodology accessible, and the visual results are well-organized to support the findings.
4. Significance: The focus on attribute-level OOD sensitivity is a valuable angle, highlighting limitations in current methods that are often overlooked.

**Weaknesses:**

1. Ecological Validity: The reliance on synthetic datasets like SHAPES and CHARS, which deviate from real-world complexity, limits the broader relevance of the findings. Real-world OOD scenarios involve more intricate and varied distributions that this controlled setup does not capture.

2. Inadequate Related works: The related work section lacks coverage of recent OOD detection benchmarks, notably missing relevant studies such as Zhao et al., 2022[1], which introduce benchmarks closer to real-world data; In the method part, lacks some state-of-the-art methods[2].This weakens the contextualization of the paper’s contributions.

3. Generalizability: Due to the simplified nature of the synthetic datasets, it remains unclear how these findings translate to actual OOD detection in high-stakes fields such as autonomous driving or healthcare.

[1] Zhao B, Yu S, Ma W, et al. Ood-cv: A benchmark for robustness to out-of-distribution shifts of individual nuisances in natural images[C]//European conference on computer vision. Cham: Springer Nature Switzerland, 2022: 163-180.
[2] Li J, Chen P, He Z, et al. Rethinking out-of-distribution (ood) detection: Masked image modeling is all you need[C]//Proceedings of the IEEE/CVF conference on computer vision and pattern recognition. 2023: 11578-11589.

**Questions:**

1. Could the authors clarify how these synthetic datasets contribute to understanding OOD detection in real-world applications?
2. Are there any plans to validate findings from SHAPES and CHARS against more complex, real-world datasets?

**Details Of Ethics Concerns:**

No more concerns

---

### Official Review · Reviewer_Fm3X · 2024-11-03

**Soundness:** 2
**Presentation:** 3
**Contribution:** 2
**Rating:** 5
**Confidence:** 3

**Summary:**

In this paper, the authors argue that existing OOD detection benchmarks do not provide precise control over individual attributes, thus lacking the ability to study the impact of individual attributes on OOD detection performance. To address this gap, the authors propose two synthetic datasets (i.e., SHAPES and CHARS) to explore OOD detection under controlled and fine-grained distribution shifts. The authors systematically evaluate 13 OOD detection methods with 10 architectures on the proposed two datasets, revealing how specific attribute shifts and image corruptions affect OOD scores.

**Strengths:**

-	The paper is generally well-written and easy to understand.
-	The authors evaluate a broad range of OOD detection methods with different architectures. The experiments are extensive.

**Weaknesses:**

-	Although the authors mention several times that existing OOD detection methods are evaluated on simple, not realistic, and small-scale datasets. It seems that the proposed synthetic datasets still face these issues. The rendered datasets are also simple and not realistic (only contain some geometric primitives and alphanumeric characters) compared to real-world images. From my perspective, creating a more realistic real-world dataset that contains general objects and scenes with more precise control over individual attribute shifts is more meaningful and impactful than the toy datasets.
-	The authors only consider two attribute shifts (i.e., color and class), which are somewhat limited. Actually, there could be many other attributes (e.g., shape and size). What if more attributes are considered and evaluated in the proposed benchmark?

**Questions:**

See the questions mentioned above. I am concerned about the actual usefulness of the toy datasets as well as limited attribute shifts. Given the current status of the paper, I am leaning towards borderline reject and hope the authors could address my concerns during the rebuttal.

---

### Official Review · Reviewer_M5v7 · 2024-11-06

**Soundness:** 3
**Presentation:** 2
**Contribution:** 1
**Rating:** 3
**Confidence:** 4

**Summary:**

Out-of-distribution (OOD) detection is essential for ensuring the reliability of machine learning models, especially in applications like healthcare and security, where high-stakes decisions rely on accurate predictions. To address the limitations of current OOD benchmarks, this study introduces two synthetic datasets, SHAPES and CHARS, designed to explore OOD detection under controlled attribute shifts. These datasets vary attributes like color, size, and rotation to systematically analyze the impact of individual attribute changes on OOD detection performance, offering insights into where current methods succeed or fail and assessing their resilience to noisy data.

**Strengths:**

- OOD problem/generalization ability is an important topic

**Weaknesses:**

- There has already been many established OOD datasets long ago, much more complex than the ones proposed here. For example, - digits: SVHN; ImageNet OOD version: ObjectNet (https://objectnet.dev/). I do not see why the proposed dataset, being so simple, would be useful or if they will be helpful in the field.
- Due to the simple setting, the experimental results can hardly generalize to OOD for real-world data, like ObjectNet.

**Questions:**

Problem setup: Line 126-137, is there a need to explain the concept in a large block of equations, especially the concept is trivial.

---

### Note · Authors · 2024-11-13

**Comment:**

After carefully reviewing the comments, we’ve decided to withdraw the paper to focus on further developing and refining our idea. The feedback has been invaluable, and we sincerely appreciate the reviewers’ time and effort. Thank you for your support in helping us improve this work.

**Withdrawal Confirmation:**

I have read and agree with the venue's withdrawal policy on behalf of myself and my co-authors.